# Behavioural diversity of bonobo prey preference as a potential cultural trait

**Liran Samuni[1,2]\*, Franziska Wegdell[3], Martin Surbeck[1,2,3]**

[1]Harvard University, Department of Human Evolutionary Biology, Cambridge, United States; [2]Max Planck Institute of Evolutionary Anthropology, Leipzig, Germany; [3]Bonobo Conservation Initiative, Washington, United States

**Abstract** The importance of cultural processes to behavioural diversity in our closest living relatives is central to revealing the evolutionary origins of human culture. However, the bonobo is often overlooked as a candidate model. Further, a prominent critique to many examples of proposed animal cultures is premature exclusion of environmental confounds known to shape behavioural phenotypes. We addressed these gaps by investigating variation in prey preference between neighbouring bonobo groups that associate and overlap space use. We find group preference for duiker or anomalure hunting otherwise unexplained by variation in spatial usage, seasonality, or hunting party size, composition, and cohesion. Our findings demonstrate that group-specific behaviours emerge independently of the local ecology, indicating that hunting techniques in bonobos may be culturally transmitted. The tolerant intergroup relations of bonobos offer an ideal context to explore drivers of behavioural phenotypes, the essential investigations for phylogenetic constructs of the evolutionary origins of culture.

## Introduction

Humans and other social animals exhibit a diversity of behavioural phenotypes attributed to genetic or social (i.e., cultural) evolutionary processes, and their combination, influenced by the environment (*Allen, 2019*; *van Schaik et al., 2003*; *Whitehead et al., 2019*; *Whiten, 2017*). While culture is identified as a pivotal selective process in human evolution (*Boyd and Richerson, 1995*; *Whitehead et al., 2019*), its relative contribution to shaping the behavioural diversity observed in non-human animals, including our closest living relatives, remains debated. For instance, in comparison to the other great ape species, little is known about potential cultural traits in bonobos (*Pan paniscus*) (*Whiten, 2017*), thereby limiting phylogenetic comparisons.

Culture is defined as group-specific behavioural patterns acquired through social learning (*Laland and Janik, 2006*). There is ample evidence that some foraging techniques are socially learned (e.g., primates [*Whiten and van de Waal, 2018*; cetaceans [*Mann et al., 2012*; carnivores [*Thornton and Raihani, 2008*]) and therefore represent good candidates for cultural traits. However, to distinguish whether social processes contribute to the emergence of behavioural phenotypes, it is essential to quantify ecological variation and account for its influence on behaviour expression, a challenging endeavour in wild settings. Few studies have attempted to limit potential ecological confounders by investigating behavioural diversity between neighbouring groups (*Luncz and Boesch, 2014*; *Pascual-Garrido, 2019*; *van de Waal, 2018*). Nonetheless, in the absence of between-group range overlap, fine-scale ecological variation specific to the locations where behavioural phenotypes are expressed cannot be excluded.

Our closest living relatives, bonobos and chimpanzees, hunt a variety of species across groups and populations (*Gilby et al., 2015*; *Hobaiter et al., 2017*; *Hohmann and Fruth, 2008*; *Sakamaki et al., 2016*; *Samuni et al., 2018*; *Wakefield et al., 2019*). However, it remains unclear whether this diversity is independent of large or even small-scale ecological variation in the

**\*For correspondence:**
lsamuni@fas.harvard.edu

**Competing interests:** The authors declare that no competing interests exist.

**eLife digest** No human culture is quite like the next. Societies around the world show exceptional variety in their social norms, beliefs, customs, language and, of course, food. However, the origins of human culture still remain elusive.

Studying humans' closest living relatives, the great apes, is one way to explore how human culture first appeared. Chimpanzees are often studied for this purpose, but other great apes, such as bonobos, are often overlooked. Yet bonobos are less territorial and more tolerant to others than chimpanzees, with different bonobo groups sharing feeding spots and hunting grounds. These traits actually make bonobos an ideal animal for investigating whether differences in group behaviour, such as feeding habits, are distinct cultural trends or just a result of their surrounding environments.

With this in mind, Samuni et al. studied the hunting and feeding patterns of two groups of wild bonobos in the Kokolopori Bonobo Reserve in the Democratic Republic of Congo. The two groups share approximately 65% of their home territory, allowing Samuni et al. to examine whether any differences in hunting preferences persisted when the two groups looked for prey in the same environment. The analysis would reveal whether social factors or environmental conditions influenced the hunting and feeding habits of each group.

Samuni et al. found the first bonobo group specialized in hunting duiker, a type of antelope, whereas the second group preferred to hunt tree-gliding rodents. However, the location and timing of the bonobo's hunts did not determine which types of prey they hunted. Across their territory, and regardless of group size or the dynamics between males and females, the groups continued to hunt their preferred prey. This means ecology alone cannot explain bonobo feeding habits and instead, the findings provide a strong indication for cultural variation between the two groups.

Since social learning is a part of cultural development, the next challenge will be to determine if and how these group hunting preferences are learned by young bonobos in their social group. For now, these findings provide a glimpse into the emergence of group culture.

distribution of prey species (*Hobaiter et al., 2017*; *Sakamaki et al., 2016*). Accounting for potential small-scale local ecological drivers is methodologically challenging in chimpanzees, a territorial species (*Mitani et al., 2010*; *Samuni et al., 2017*) where each group predominantly occupies unique non-overlapping areas. In contrast, the tolerant intergroup relations of bonobos (*Furuichi, 2020*) permit a context in which different behaviours are expressed by individuals of different groups in the same place and at the same time. Here, we investigate variation in bonobo predation patterns of two groups (Ekalakala and Kokoalongo) at the Kokolopori Bonobo Reserve. The groups share an extensive home range overlap (65% kernel overlap; *Figure 1*,A,B,C) and regular gene flow, thereby reducing ecological and genetic influences as an explanatory variable for intergroup differences in behavioural expressions (*van de Waal, 2018*). Specifically, we tested whether variation in prey preference between the two bonobo groups is explained by a) environmental variables, such as area usage and seasonality, and/or b) social factors, such as the number of potential hunters, individual association pattterns, and group identity.

## Results

Between August 2016 and January 2020, we observed 59 successful captures and consumption of mammals by the bonobos, including anomalure, duiker, and squirrel species (*Table 1*; *Figure 1—figure supplement 1*; *Video 1*). Starting July 2019, we also collected data on unsuccessful hunts, and documented 11 hunt attempts on duiker and anomalure (duiker- $N_{Ekalakala}$ = 2, $N_{Kokoalongo}$ = 2; anomalure- $N_{Ekalakala}$ = 4, $N_{Kokoalongo}$ = 3). Overall, we observed all Ekalakala and 84% of Kokoalongo adult group members (100% if considering only individuals that were present for the entire study period) participating in hunts.

Most anomalure and duiker hunts occurred within overlapping ranging areas (94% of anomalure and 83% of duiker hunts), compared to only 46% of squirrel hunts (*Figure 1*,A,B,C). The groups engaged in frequent and prolonged intergroup associations (31% of observation days), and nine of the hunts (five duiker, three anomalure, one squirrel) occurred during intergroup encounters and at times involved between-group meat sharing. Although 45% of the Kokoalongo duiker hunts

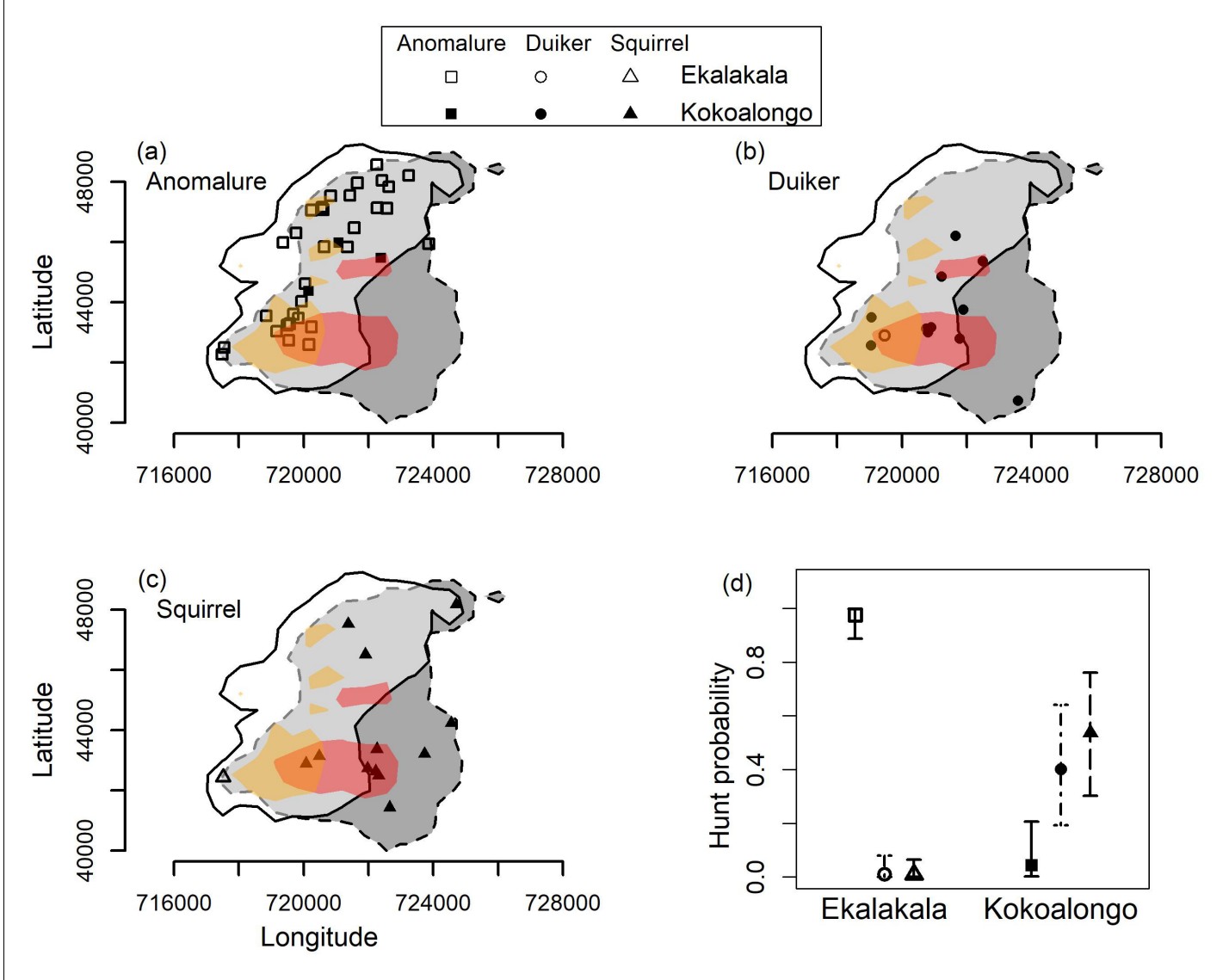

**Figure 1.** Predation patterns in Kokolopori bonobos. Hunting locations (*Figure 1—source data 1*) of the three prey types: (**a**) anomalure (square), (**b**) duiker (circle), and (**c**) squirrel (triangle) in relation to the 95% Kernel usage area of Ekalakala (white polygon with solid border) and Kokoalongo (dark grey polygon with dashed border) and 50% Kernel usage area (Ekalakala in yellow, Kokoalongo in red). The overlapping 95% kernel area between Ekalakala and Kokoalongo is depicted in light grey. Also depicted are (**d**) the predicted hunt probabilities of the different prey types between Ekalakala and Kokoalongo as obtained from the BR model (*Figure 1—source data 2*).

The online version of this article includes the following source data and figure supplement(s) for figure 1:

**Source data 1.** Hunt locations of the different prey types.
**Source data 2.** Predicted hunt probabilities of the different prey types.
**Figure supplement 1.** Prey species categories hunted by the Kokolopori bonobos.

occurred during encounters, very few to none (mean = 1.4) of the Ekalakala individuals were present during these hunts, and none participated (*Supplementary file 1*). Due to the cohesiveness of bonobo groups (*Hohmann and Fruth, 2002*), the conspicuous nature of anomalure and duiker hunting (e.g., distress calls of duikers), and since the acquisition of meat often attracts individuals to hunting areas (*Samuni et al., 2018*), we are confident that we observed most anomalure and duiker feeding events. However, as the hunting and feeding of squirrel is often quiet and solitary and since hunting is frequently detected only post capture, we are likely to have underestimated this type of hunting.

**Table 1.** Successful hunts in Ekalakala and Kokoalongo between August 2016-Jan 2020.

| Group | Anomalure[*] | Duiker[†] | Squirrel[‡] |
|---|---|---|---|
| Ekalakala | 31 | 1 | 1 |
| Kokoalongo | 3 | 11 | 12 |

\* *Anomalurus derbianus, Anomalurus beecrofti.*

† *Philantomba monticola, Cephalophus castaneus.*

‡ *Funisciurus congicus.*

Kokoalongo bonobos were more likely to capture duiker (estimate = 4.56, $CI_{95\%}$ = [1.93, 8.03]; *Figure 1D*, *Table 2*) and squirrel species (estimate = 4.99, $CI_{95\%}$ = [2.34, 8.21]), and were less likely to capture anomalure species in comparison with Ekalakala. The same pattern persisted during inter-group encounters (once we observed anomalure captured by a Kokoalongo female after a hunt by Ekalakala individuals; *Supplementary file 1*). We found that prey preferences were independent from potential local spatial and temporal ecological variation. Overall, more than 80% of all hunts occurred in overlapping areas (95% kernel), and neither utilization differences of specific hunt locations (reflecting varying opportunities to encounter prey species) nor potential annual seasonal variation strongly affected phenotypic variation in prey types captured (*Table 2*). Variation in prey preference can also arise from between-group difference in sizes of female or male association parties, association tendencies amongst party members, or presence of certain specialized hunters. However, the number of adult females or males present during hunts (i.e., available hunters) and the average dyadic association between them had no strong effect on prey outcome (*Table 2*). Further, we observed 17 different individuals (five males and 12 females) catching prey, encompassing 72% of Ekalakala and 40% of Kokoalongo group members (see *Supplementary file 1* for the distributions of catchers). These percentages are likely an underestimation of the overall number of individuals who captured the prey, as their identity was not recorded for 40% of all hunts. Finally, our results are likely independent from genetic variation, as low genetic differentiation is expected (*Schubert et al., 2011*) mainly due to regular gene flow attributed to female migration between Ekalakala and Kokoalongo.

## Discussion

We found that bonobo groups that utilize overlapping home ranges and regularly socialize and forage together show group-specific prey acquisition patterns. These group-specific patterns appear independent of genetic and small-scale ecological variation, seasonality, size of hunting parties, or party cohesiveness. The exclusion of these confounders indicates that other drivers of behavioural variation act as mechanisms in prey selection.

Observed differences in prey preferences may arise if different techniques are required to locate and capture them. Duiker and squirrel hunting are either strictly terrestrial (duiker) or arboreal (squirrel) activities, which appear opportunistic and commonly involved a single individual hunter (more so for squirrel hunting). Conversely, anomalure hunting required the engagement of several group members, during which the bonobos employed both terrestrial and arboreal positions. While at this stage it is unclear if hunting techniques in bonobos require time to acquire or involve social learning processes, specialized hunting techniques may be at the basis of the observed group differences.

Prey species preference may additionally reflect differences in prey palatability between groups. Although between-group meat sharing of duiker and anomalure may contradict the idea of group specific meat preference, the costs and benefits associated with hunting relative to

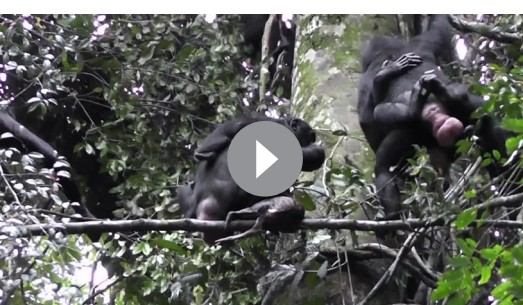

**Video 1.** Duiker and anomalure hunting by Kokolopori bonobos.

https://elifesciences.org/articles/59191#video1

**Table 2.** Bayesian Regression model results of the effect of group identity, number of available hunters and ecological variation on prey species captured ([1]anomalure and [2]Ekalakala as reference categories).

All numeric predictor variables were standardized to mean = 0 and sd = 1.

| Coded level | Term | Estimate | SE | 95% CI |
|---|---|---|---|---|
| Duiker[1] | Intercept | −3.25 | 1.04 | −5.50,−1.51 |
| | **Group (Kokoalongo[2])** | **4.56** | **1.57** | **1.93, 8.03** |
| | Available male hunters | 0.43 | 0.77 | −1.05, 1.99 |
| | Available female hunters | 0.42 | 0.75 | −1.07, 1.90 |
| | Association | −0.77 | 0.74 | −2.30, 0.68 |
| | Usage difference | 0.39 | 0.55 | −0.63, 1.52 |
| | Sine of Date | 1.24 | 0.82 | −0.33, 2.89 |
| | Cosine of Date | 0.00 | 0.84 | −1.68, 1.63 |
| Squirrel[1] | Intercept | −3.32 | 1.03 | −5.61,−1.52 |
| | **Group (Kokoalongo[2])** | **4.99** | **1.51** | **2.34, 8.21** |
| | Available male hunters | 0.50 | 0.80 | −1.09, 2.11 |
| | Available female hunters | −0.18 | 0.77 | −1.66, 1.30 |
| | Association | −0.61 | 0.73 | −2.06, 0.79 |
| | Usage difference | 0.71 | 0.55 | −0.32, 1.89 |
| | Sine of Date | 1.03 | 0.79 | −0.47, 2.67 |
| | Cosine of Date | 0.36 | 0.81 | −1.21, 1.92 |

begging potentially alter consumption decisions. As hunting behaviour is associated with energetic costs, the benefit of capturing favourable prey may persuade hunt decision making. Conversely, once prey is captured, the costs associated with begging are minimal relative to hunting, thereby largely resetting the cost-to-benefit ratio behind foraging decisions. Thus, while palatability may dictate which prey species to pursue, it is expected to have a lesser impact on begging decisions.

The 'impact hunter' hypothesis (*Gilby et al., 2015*) could offer an alternative explanation for prey preference variation, proposing that certain individuals encourage social hunts by assuming hunt initiation costs. However, as this hypothesis addresses social hunt occurrence, it could explain the prevalence of social hunts like anomalure but cannot explain why duiker and squirrel hunting (opportunistic and largely solitary) are nearly absent in Ekalakala. Further, we observed many individuals participating in hunts and capturing prey and prey outcome was independent of the number of male or female hunters. Thus, patterns in our data indicate that we indeed document group, instead of individual, tendencies.

In the absence of ecological, genetic, or ingroup social dynamic explanations of prey acquisition, the observed group-specific differences may be cultural. Under this assumption, it is puzzling how such group differences would evolve and persist even when prolonged associations between Ekalakala and Kokoalongo should potentially promote intergroup social learning opportunities. Tolerance, at a degree that facilitates social learning in its various forms, is fundamental in converting innovations into transmitted traditions (*Whiten and van de Waal, 2018*). To improve 'learning' gains, social learners should be selective in the timing of observations and their choice of 'models' from whom to learn (*Boyd and Richerson, 1995*). Although the two groups associate for extended periods their intergroup relations are complex and unpredictable, characterized by a mixture of affiliative and agonistic exchanges, frequent fission-fusions and heightened arousal. Unpredictability of intergroup interactions is thus expected to hamper intergroup learning opportunities of certain skills which may require extensive time and effort to acquire (e.g., hunting techniques). Following group psychology predictions of ingroup bias and favouritism (*Brewer, 1993*), outgroup members may as well be less appealing 'models' for learning. Together, inconsistent intergroup relations and in-group bias may explain how group-specific prey preferences persist despite numerous intergroup learning opportunities. A by-product of divergent hunting techniques is reduced intergroup competition, which is

likely adaptive, especially when groups share ranging zones. Thus, group-specific prey preferences in bonobos may have evolved as a form of microlevel niche differentiation that alleviates feeding competition.

Investigating the potential impact of culture on behavioural diversity in non-human animals is challenging due to the difficulties of estimating and accounting for local ecological variation as a driver of behavioural diversity. Challenges may even arise when behavioural variation appears between groups that occupy nearby but non-overlapping ranging areas. Bonobo social groups' regular overlap in ranging area and tolerant interactions, offer fertile ground in which to explore whether variation in behavioural expressions occurs independently of spatial and temporal use of specific habitat locations. Here, by accounting and largely excluding potential local ecological variation, we provide strong indication for culturally transmitted subsistence hunting techniques in bonobos, informing on the evolution of behavioural diversity.

## Materials and methods

### Study site and data collection

We investigated behavioural diversity between two fully habituated bonobo groups (Ekalakala and Kokoalongo, followed since 2007) at the Kokolopori Bonobo Reserve, Democratic Republic of Congo (N 0.41716°, E 22.97552°; [*Surbeck et al., 2017a*]). We conducted full day party follows of the bonobo groups (1102 and 931 observation days in Ekalakala and Kokoalongo, respectively) and documented all occurrence hunting behaviour (here defined as capture of mammalian prey). All prey types were captured across most months, and both during the dry (June-August and December-February) and wet (March-May and September-November) seasons. Hunt participants were almost exclusively adult (>10 years) individuals, and both sexes were observed to participate. Adult group sizes fluctuated during the study between 9–11 adult individuals in Ekalakala and 16–24 adult individuals in Kokoalongo due to several deaths and migration events (*Supplementary file 2*).

### Home range utilization distribution

We recorded data on party locations at one-minute intervals using a GPS (Garmin 62). We constructed home range utilization distributions of the bonobo groups using kernel density estimates (*Worton, 1989*). The home range (95% kernel) of the two groups between August 2016 and December 2019 was: Ekalakala – 35 km$^2$, Kokoalongo – 40 km$^2$, and the overlapping area encompassed 64% and 66% of the home ranges of Ekalakala and Kokoalongo, respectively.

Habitat structure and spatial distribution of prey species have been used as explanations for variation in hunting behaviours (*Hobaiter et al., 2017*; *Sakamaki et al., 2016*). However, as our data originate from two groups with extensive home range overlap, the explanatory power of these drivers is minimized. Nonetheless, we can evaluate intra-range variation in local ecology by accounting for relative home range usage across the groups. To do so, we assigned each hunt with two kernel usage values, one constituting the kernel usage of the group that hunted (*hunt group*) and the other constituting the kernel usage of the group that did not hunt (*other group*). We used the values to calculate a score of 'usage difference' (i.e., *other group - hunt group*; ranging between −50 and 86; mean ± sd: 20.19 ± 26.10). Higher scores reflected an area that is more predominantly used by the group that hunted.

### Association patterns

We recorded the cumulative adult party composition at 30 min intervals and marked individuals observed during the hunt scan as potential hunters. Whenever a party composition scan collected either immediately before or during a hunt included individuals of both groups (representing between-group spatial proximity), that hunt was marked as occurring during an intergroup encounter. This approach categorized two hunts as intergroup hunts although members of only one group were present, but accounts for the likelihood that the other group is nearby.

We used these party scans to calculate dyadic association values for each dyad and year, using the following equation: SRI = $P_{AB}/(P_A + P_B - P_{AB})$ (*Surbeck et al., 2017b*). $P_A$ and $P_B$ represent the number of scans A or B were present, and $P_{AB}$ represents the number of scans both A and B were

present. For every hunt, we then calculated the average dyadic association of the hunting party as a proxy of group social cohesion, which may affect the likelihood to capture prey.

## Statistical analysis

We applied a Bayesian Regression model with prey type as a categorical response and logit link function to examine the influence of environmental (area usage and seasonality) and social (group identity, presence of potential hunters, and social cohesion) factors on prey preference expression. We fitted the model in R (version 3.6.1 [*R Development Core Team, 2016*]) using the function *brm* of the R package 'brms' (*Bürkner, 2017*) and weakly informative t-distributed priors (*Lemoine, 2019*). As predictors, we included the following environmental factors: a) 'usage difference' score as described above, and b) a seasonal temporal term, by including the sine and cosine of the Julian dates of the hunts converted into a continuous circular variable (*Stolwijk et al., 1999*). The sine and cosine predictors allow for the modelling of a wave like periodic pattern of peaks and valleys, thereby representing potential seasonal oscillations in hunt dates. Additionally, we included the following social factors: a) group identity of the individual who caught the prey, b) female and male party sizes (mean ± sd: Ekalakala - 7.19 ± 1.47; Kokoalongo – 7.05 ± 3.62; encounter - 13 ± 7.4), and c) average dyadic associations of hunt party mean ± sd: Ekalakala - 0.51 ± 0.09; Kokoalongo – 0.34 ± 0.13; encounter – 0.26 ± 0.14). Note, if dietary requirements alone were to dictate hunting patterns, then we would expect a random distribution (reflecting prey species encounter probabilities) of the different prey species captured within groups instead of group-specific patterns.

We ran 2000 iterations over four MCMC chains, with a 'warm-up' period of 1000 iterations per chain leading to 4000 usable posterior samples (*Bürkner, 2017*). Visual inspection of all MCMC results revealed satisfactory Rhat values (<1.01; [*Gelman et al., 2013*]), no divergent transitions after warmup, and stationarity and convergence to a common target, suggesting that our results are stable. We report the estimate (mean of the posterior distribution) and the 95% credible intervals ($CI_{95\%}$) indicating the strength of the effects. For estimate comparability and to ease model convergence, we standardized all numeric variables to mean = 0 and sd = 1. Our model did not suffer from issues of collinearity, evaluated using Variance Inflation Factors (*Field et al., 2012*) with the R package 'car' (*Fox et al., 2020*). The data reported in this paper are available as *Source data 1*.

## Acknowledgements

We are grateful to the Bonobo Conservation Initiative, Vie Sauvage and Institut Congolais pour la Conservation de la Nature, especially Sally Coxe and Albert Lotana Lokasola, for their continuous support of this work. We thank the Ministry of Research of the Democratic Republic of the Congo for permitting the study, and the people of the villages of Bolamba, Yete, Yomboli and Yasalakose for granting access to their forest. We thank all the research assistants and local field assistants for their dedication and support in the field and for Erin Wessling and Catherine Hobaiter for their helpful comments on a previous version of this manuscript. We thank Erica Van de Waal, Detlef Weigel, and two anonymous reviewers for their valuable comments towards improving this manuscript. This work is funded by the Max Planck Society and Harvard University.

## Additional information

### Funding

| Funder | Author |
| --- | --- |
| Harvard University | Liran Samuni<br>Martin Surbeck |
| Max-Planck-Institut für Evolutionäre Anthropologie | Liran Samuni<br>Martin Surbeck |

The funders had no role in study design, data collection and interpretation, or the decision to submit the work for publication.

## Author contributions
Liran Samuni, Conceptualization, Data curation, Software, Formal analysis, Investigation, Visualization, Methodology, Writing - original draft, Writing - review and editing; Franziska Wegdell, Conceptualization, Data curation, Investigation, Writing - review and editing; Martin Surbeck, Conceptualization, Resources, Supervision, Funding acquisition, Investigation, Project administration, Writing - review and editing

## Author ORCIDs
Liran Samuni ⓘD https://orcid.org/0000-0001-7957-6050

## Ethics
Animal experimentation: The research presented here was non-invasive and approved by the Ministry of Research of the Democratic Republic of the Congo (permit N° 013/CAB.MINRST/DMK/DK/2017 to MS). This study complies with the ethics policy of the Max Planck Society and the Department of Primatology of the Max Planck Institute for Evolutionary Anthropology, Germany (https://www.eva.mpg.de/primat/ethical-guidelines.html) and the American Society of Primatologists principles for the ethical treatment of non-human primates.

## Decision letter and Author response
Decision letter https://doi.org/10.7554/eLife.59191.sa1
Author response https://doi.org/10.7554/eLife.59191.sa2

## Additional files

### Supplementary files
• Source data 1. Dataset used in the statistical analysis.

• Supplementary file 1. Successful hunt cases on anomalure, duiker, and squirrel species documented between August 2016 and January 2020 in Ekalakala (EKK) and Kokoalongo (KKL). The identity and sex (M = male, F = Female) of the individual that caught the prey is noted whenever information was available (NA = unknown). The party composition columns depict the different group members that were present during the hunt scan. In bold presented are the individuals that participated in the hunt (i.e., chased prey) when known. Note, we have at times likely underestimated the number of hunters.

• Supplementary file 2. Demography of the adult individuals of the Ekalakala and Kokoalongo groups, 2016–2020 (M = male, F = Female).

• Transparent reporting form

### Data availability
The data reported in this paper are available as Source Data 1 and supplementary files. Source data files have been provided for Figure 1.

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
