## [Decision Letter]

**Acceptance summary:**

The present paper is important because it reports behavioural variation between two overlapping groups of some of our closest relatives: bonobos. The authors provide a very interesting potential cultural trait in a relatively under-reported domain, predatory behaviour. Moreover, this research is able to provide compelling evidence of this behavioural variation due to the large overlap in the two groups territories, which convincingly sets aside the alternative explanation of ecological foundations for the observed variation.

**Decision letter after peer review:**

Thank you for submitting your article "Behavioural diversity of bonobo prey preference as a potential cultural trait" for consideration by *eLife*. Your article has been reviewed by three peer reviewers, including Erica Van de Waal as the Reviewing Editor and Reviewer #1, and the evaluation has been overseen by Detlef Weigel as the Senior Editor.

The reviewers have discussed the reviews with one another and the Reviewing Editor has drafted this decision to help you prepare a revised submission.

Summary:

This is a very interesting report of apparent hunting differences in terms of prey preference in two neighbouring/overlapping bonobo groups. The paper is well-written and clear, as are the figures, and the methods and statistical analyses are rigorous, such that the authors' principal conclusions are well supported. While all reviewers enjoyed reading the article and were generally positive about the data (particularly in that they address the 'ecological' argument usually brought up by animal culture critics), some of the claims seem too strong and should be better substantiated by data presented directly in the main text.

We ask the authors to revise their paper following the comments here below. If the authors can address our concerns, we agree that the paper should be published in *eLife*, but the claims on culture should be toned down in line with what the data show.

1) Reviewer 3 has in particular a strong concern regarding the duiker data, which he fails to understand with the way data are presented in the manuscript at the moment (the raw data do not appear to be in mat sup either). This is crucial, because what the authors present as 'cultural differences' may also be just one group having a certain preference for a given prey (anomalures: whether this in itself is enough to claim cultural differences can be debated, enough for some authors, and not for others). In addition, there is no evidence at present that this group difference results from social learning.

2) Group composition (Sex differences and ID of hunters): we disagree with authority arguments of the type "because we do not expect it to have a difference on prey acquisition". Where does this come from, especially as it is well documented in the literature that there are sex differences regarding hunting in bonobos, which is often assumed to be female-led? It seems like it would be an important factor, in fact even more so for the identity of the main 'driver' of the hunt (or the Gilby 'impact hunter'). The results for the catcher also suggest that sex would have been a good factor to include in the analysis. What if females have a preference for anomalures and males for duikers? If there are more females in group A than group B, this can lead to apparent cultural differences because a given sex dominates the hunt in one group compared to the other. The authors need to provide more data to address this. In particular, it would be great to have a table that at least shows the different IDs of the 'catchers', as there is already a sex bias appearing with females more likely to catch the prey than males (which group do they belong to? Are the sex split between the two communities?). This will possibly already address the issue of personal preferences.

General comment for the Discussion: you found that the smaller group was mainly capturing anomalures, could it be not explained by the fact that smaller group are more cohesive and coordinated? Even if you checked for party size could you control for the composition of the party size? If in the larger group individuals are less bonded and stay more in the same party without successful anomalure hunters they might be less likely to hunt them? Whereas in the smaller group the individuals might be spending more time with all other group members?

3) Intergroup hunts: There is a lack of clarity on which hunt was an intergroup hunt in Table 1 and it is particularly concerning regarding the duiker hunt data. Out of 59 successful hunts, 9 were considered intergroup hunts. If 5 of the successful duiker hunts (as in, including meat sharing) involve members of both groups, why are they classified as 1 Ekalakala and 11 Kokoalongo hunts? Is it because the 'catcher' is from one of the groups only? But then if the ones who consume the food are from both groups, it's hard to argue that there would be 'culinary' cultural differences between the two groups (at best, the cultural difference is in the likelihood to engage in a certain type of hunting for a given prey). Thus shouldn't these hunts be taken out from the group analysis? The authors only state that 'the same pattern persisted during intergroup encounters'. The problem then is that whether these particular hunts are shared between both groups or removed, the possible duiker difference probably becomes minimal and largely irrelevant for lack of data. If that is the case, based on the remaining data, the authors can probably identify a strong preference in one group (Ekalakala for anomalure), but it gets harder to argue for cultural differences (particularly because there is no evidence provided for social learning processes).

4) Culture/social learning claims: The Discussion starts with the authors saying that the "findings demonstrate that social processes….". That is not true. The Materials and methods and Results do not actually test for any type of social influence or social learning on hunting. This sentence should thus be modified and in general, the discussion on specific kinds of social learning mechanism (emulation etc…) toned down or removed because there is nothing in the manuscript that can help with this question. It would be really nice to see (pending confirmation), that one community is more opportunistic in their hunting (duiker + squirrel) compared to the other that seems to go more collectively. However, the authors cannot take evidence for social learning for collective hunting from the literature (here, findings in dolphin hunting and a more general review in primates including foraging) as evidence for their own findings, which are quite distinct, starting with the study species. The authors' finding is that there is a clear difference in terms of (at least one) prey preference between two neighbouring/overlapping bonobo groups. Whether this is cultural can then be discussed in light of what the authors have shown (e.g. no effect of ecology). Hence it would be more correct to say: we have observed this difference, can it be cultural? In contrast, the last two paragraphs are good to keep, once the assumption that the difference is cultural is met.

5) Unsuccessful hunts: you need to clarify your hypothesis about the unsuccessful hunts. Is the argument about the unsuccessful hunts (Discussion, second paragraph) really valid if it is based on only 6 months of data (compared to the hunt count that spans several years)? Also, while reading the Results, it seemed that it was the entire dataset for unsuccessful hunts; but that does not seem to be the case in fact. It should probably be introduced differently to avoid confusion then. In general, this specific paragraph in the Discussion needs to be reworked as well if it aims to address the possible "Gilby argument". It can only be properly addressed by offering a list of IDs that shows diversity in individual hunters/catchers, and not by making group pattern remarks.

Revisions expected in follow-up work:

You need to add a table (for the hunts where you were able to record the identity of all participants) with the list of the different individuals that have been observed hunting on all three types of prey in both group, with their individual characteristics (such as age, sex, rank, relatedness of participants), and if the hunt with successful or not.

---

## [Author Response]

Summary:This is a very interesting report of apparent hunting differences in terms of prey preference in two neighbouring/overlapping bonobo groups. The paper is well-written and clear, as are the figures, and the methods and statistical analyses are rigorous, such that the authors' principal conclusions are well supported. While all reviewers enjoyed reading the article and were generally positive about the data (particularly in that they address the 'ecological' argument usually brought up by animal culture critics), some of the claims seem too strong and should be better substantiated by data presented directly in the main text.

We thank the anonymous reviewers, Reviewing Editor Erica Van de Waal, and Senior Editor Delef Weigel for their valuable and constructive comments which have helped to improve our manuscript. We have addressed all their comments by incorporating additional predictors into the model (i.e., the number of males or female party members, association patterns of party members), adding tables, figure supplement, and video, and by toning down cultural claims.

Revisions for this paper:We ask the authors to revise their paper following the comments here below. If the authors can address our concerns, we agree that the paper should be published in eLife, but the claims on culture should be toned down in line with what the data show.1) Reviewer 3 has in particular a strong concern regarding the duiker data, which he fails to understand with the way data are presented in the manuscript at the moment (the raw data do not appear to be in mat sup either). This is crucial, because what the authors present as 'cultural differences' may also be just one group having a certain preference for a given prey (anomalures: whether this in itself is enough to claim cultural differences can be debated, enough for some authors, and not for others). In addition, there is no evidence at present that this group difference results from social learning.

This is a valuable point which we failed to clarify in the initial submission. As suggested, we have added a table of all documented successful hunts (Supplementary file 1), together with information of party composition by group, and when available, the identity and sex of the individual who captured the prey and the identity of individuals that participated in the hunt. We also provide more information on duiker hunts in the manuscript and toned-down discussions of social learning.

2) Group composition (Sex differences and ID of hunters): we disagree with authority arguments of the type "because we do not expect it to have a difference on prey acquisition". Where does this come from, especially as it is well documented in the literature that there are sex differences regarding hunting in bonobos, which is often assumed to be female-led? It seems like it would be an important factor, in fact even more so for the identity of the main 'driver' of the hunt (or the Gilby 'impact hunter'). The results for the catcher also suggest that sex would have been a good factor to include in the analysis. What if females have a preference for anomalures and males for duikers? If there are more females in group A than group B, this can lead to apparent cultural differences because a given sex dominates the hunt in one group compared to the other. The authors need to provide more data to address this. In particular, it would be great to have a table that at least shows the different IDs of the 'catchers', as there is already a sex bias appearing with females more likely to catch the prey than males (which group do they belong to? Are the sex split between the two communities?). This will possibly already address the issue of personal preferences.

The reviewers are correct to suggest that sex differences may drive prey acquisition patterns, which may explain group specific prey preference. We have now incorporated additional information on the group identity and sex of the individuals that captured the prey within the main text and as supplementary tables (Supplementary files 1-2). Further, to account for potential sex-differences in prey acquisition in the statistical analysis, we now include the number of available hunters as two separate covariate predictors, the number of male or female party members. Neither of the two predictors showed a clear effect on the response nor altered the overall model results.

General comment for the Discussion: you found that the smaller group was mainly capturing anomalures, could it be not explained by the fact that smaller group are more cohesive and coordinated? Even if you checked for party size could you control for the composition of the party size? If in the larger group individuals are less bonded and stay more in the same party without successful anomalure hunters they might be less likely to hunt them? Whereas in the smaller group the individuals might be spending more time with all other group members?

This is a nice suggestion. To account for the idea that social cohesion may drive hunt success of different species, we have incorporated into the model the average dyadic association values of hunt party members (i.e., adult individuals present during the hunt). The average dyadic association value serves as a proxy for social cohesion, such that higher values represent parties of individuals that associate more frequently overall. By including this predictor, we are accounting for some of the variation in prey type that can potentially be explained by social familiarity (see Results).

3) Intergroup hunts: There is a lack of clarity on which hunt was an intergroup hunt in Table 1 and it is particularly concerning regarding the duiker hunt data. Out of 59 successful hunts, 9 were considered intergroup hunts. If 5 of the successful duiker hunts (as in, including meat sharing) involve members of both groups, why are they classified as 1 Ekalakala and 11 Kokoalongo hunts? Is it because the 'catcher' is from one of the groups only? But then if the ones who consume the food are from both groups, it's hard to argue that there would be 'culinary' cultural differences between the two groups (at best, the cultural difference is in the likelihood to engage in a certain type of hunting for a given prey). Thus shouldn't these hunts be taken out from the group analysis? The authors only state that 'the same pattern persisted during intergroup encounters'. The problem then is that whether these particular hunts are shared between both groups or removed, the possible duiker difference probably becomes minimal and largely irrelevant for lack of data. If that is the case, based on the remaining data, the authors can probably identify a strong preference in one group (Ekalakala for anomalure), but it gets harder to argue for cultural differences (particularly because there is no evidence provided for social learning processes).

Per the reviewers’ comment, we understand that there was unclarity regarding definitions of intergroup encounters, the group identity associated with the hunt, and the overall duiker hunt data in initial submission.

Hunts were classified as Ekalakala or Kokoalongo depending on the identity of the individuals that participated in the hunt, and in cases when both groups participated, the identity of the individual that caught the prey defined the group identity (see Discussion).

We defined a hunt as occurring during an intergroup encounter whenever individuals of both groups were observed in the same party, either during the hunt scan or during the immediate scan prior to the hunt. This was done to account for the high likelihood that the other group is nearby when observed in temporal and spatial vicinity to the hunt. We have added the intergroup encounter definition in the Results. By this approach, two of the nine hunts defined as intergroup encounters included only Kokoalongo individuals (two duiker hunts). We believe that this definition is reliable as in one of the two cases, an Ekalakala female immediately joined the Kokoalongo party once duiker distress calls were heard after the successful capture, and received a share of the meat (Supplementary file 1).

Although, as the reviewer pointed out, 45% of Kokoalongo duiker hunts occurred during intergroup encounters, the average number of Ekalakala individuals present during these hunts was 1.4 and none of them participated in the hunt. Further, while Kokoalongo individuals are frequently observed to hunt adult duikers (Video 1), the only duiker hunt observed in Ekalakala involved the capture of a duiker calf from its hiding place (involving a different hunting technique than what is needed for adult duikers). We have added this information in the main text (Results) and in the legend of the additional figure (Figure 1—figure supplement 1). Taken together, we argue that the duiker data is meaningful for the observed between-group prey difference.

The reviewers also suggest that between group meat sharing observations likely refute ‘culinary’ differences between the groups. However, here we argue that prey palatability differences may still be at the basis of the observed group differences because the costs associated with hunting for meat are very different (higher) than the costs associated with begging for meat. While palatability differences may dictate hunting decisions so to maximize the cost-to-benefit ratio, once prey is already captured the costs associated with access to meat are minimized. In sum, bonobos might not initiate a hunt on a less preferred prey due to hunt costs but will beg for the meat when others have caught it. We have added this as part of the Discussion.

4) Culture/social learning claims: The Discussion starts with the authors saying that the "findings demonstrate that social processes….". That is not true. The Materials and methods and Results do not actually test for any type of social influence or social learning on hunting. This sentence should thus be modified and in general, the discussion on specific kinds of social learning mechanism (emulation etc…) toned down or removed because there is nothing in the manuscript that can help with this question. It would be really nice to see (pending confirmation), that one community is more opportunistic in their hunting (duiker + squirrel) compared to the other that seems to go more collectively. However, the authors cannot take evidence for social learning for collective hunting from the literature (here, findings in dolphin hunting and a more general review in primates including foraging) as evidence for their own findings, which are quite distinct, starting with the study species. The authors' finding is that there is a clear difference in terms of (at least one) prey preference between two neighbouring/overlapping bonobo groups. Whether this is cultural can then be discussed in light of what the authors have shown (e.g. no effect of ecology). Hence it would be more correct to say: we have observed this difference, can it be cultural? In contrast, the last two paragraphs are good to keep, once the assumption that the difference is cultural is met.

We followed the reviewers’ comment and modified claims of cultural transmission or discussion of social learning and further emphasize our findings regarding the lack of ecological effects.

5) Unsuccessful hunts: you need to clarify your hypothesis about the unsuccessful hunts. Is the argument about the unsuccessful hunts (Discussion, second paragraph) really valid if it is based on only 6 months of data (compared to the hunt count that spans several years)? Also, while reading the results, it seemed that it was the entire dataset for unsuccessful hunts; but that does not seem to be the case in fact. It should probably be introduced differently to avoid confusion then. In general, this specific paragraph in the Discussion needs to be reworked as well if it aims to address the possible "Gilby argument". It can only be properly addressed by offering a list of IDs that shows diversity in individual hunters/catchers, and not by making group pattern remarks.

Successful hunts have only been recorded starting July 2019, when LS initiated field work in Kokolopori. Thus, we unfortunately only have preliminary data regarding unsuccessful hunts. We have reworded the sentence to clarify this “Starting July 2019 we also collected data on unsuccessful hunts”. We have also decided to remove this section from the Discussion, as we agree with the reviewer that the results are preliminary and sample size is low.

Revisions expected in follow-up work:You need to add a table (for the hunts where you were able to record the identity of all participants) with the list of the different individuals that have been observed hunting on all three types of prey in both group, with their individual characteristics (such as age, sex, rank, relatedness of participants), and if the hunt with successful or not.

We have added two tables (Supplementary files 1-2). One table includes the identities of all individuals per group with information on their sex and age categories (Supplementary file 2), and a second table with information of all successful hunt cases, including the identity of individuals present during the hunt by group membership, and when available the identity and sex of the individual who caught the prey and of those that participated in the hunt (Supplementary file 1).